# Effects of the Host Plants of the Maize-Based Intercropping Systems on the Growth, Development and Preference of Fall Armyworm, *Spodoptera frugiperda* (Lepidoptera: Noctuidae)

**DOI:** 10.3390/insects15010026

**Published:** 2024-01-02

**Authors:** Wen-Cai Tao, Xue-Yan Zhang, Yue Zhang, Xiao-Yue Deng, Hui-Lai Zhang, Zhi-Hui Zhang, Qing Li, Chun-Xian Jiang

**Affiliations:** College of Agronomy, Sichuan Agricultural University, Chengdu 611130, China

**Keywords:** maize-based intercropping systems, *Spodoptera frugiperda*, growth, development, preference

## Abstract

**Simple Summary:**

In this study, we investigated the effects of the host plant of the maize-based intercropping systems, including maize, sweet potato, soybean and peanut, on the growth, development and preference of fall armyworm (FAW), *Spodoptera frugiperda* (Smith), in the laboratory. The results showed that maize and peanut were suitable for the survival and development of FAW, while sweet potato and soybean were not suitable for multigenerational reproduction. The larvae significantly preferred to feed on maize compared to the other three plants. Similarly, the female preferred to oviposit and lay more eggs on maize rather than on other plants. The intercropping plants significantly reduced the oviposition selection of FAW adults compared to maize alone. Our study also indicated that soybean reduced FAW larval fitness and had a strong deterrence effect on the host location and oviposition of FAW. This study provided a reference for further research on the pattern of FAW occurrence and damage as well as comprehensive management in the maize-based intercropping systems.

**Abstract:**

In this paper, the effects of maize and its three intercropping plants, sweet potato, soybean and peanut, on the growth and development of FAW, feeding preference of larvae, olfactory response and oviposition preference of adults were studied in the laboratory. The results showed that maize and peanut were suitable for the survival and development of FAW, while sweet potato and soybean were not suitable for multigenerational reproduction. The larvae significantly preferred to feed on maize compared to the other three plants. The olfactory response test indicated that soybean showed a strong deterrent effect against FAW adults. Furthermore, the intercropping plants reduced the host selection rate of adults compared to maize alone. In two-choice tests of the maize vs. the intercropping plants, the female adult preferred to oviposit and lay more eggs on maize rather than on the intercropping plants. The intercropping plants significantly reduced the oviposition selection of FAW adults when the combination (maize + intercropping plant), especially soybean and sweet potato, was compared to maize alone. These may be the reasons for why the maize–soybean intercropping system reduced FAW damage in the field. We also speculated that the maize–sweet potato system may also reduce the FAW damage. This study provided a theoretical basis for the comprehensive management of FAW by utilizing an intercropping system.

## 1. Introduction

The fall armyworm (FAW), *Spodoptera frugiperda* (Smith), native to tropical and subtropical regions of the Americas [1], is a major migratory agricultural pest worldwide. Due to its strong migratory ability [2], high fecundity [3], a wide range of hosts [4] and polyphagous and voracious feeding [5], FAW has spread rapidly in recent years and caused huge economic losses to crops [6,7,8]. FAW invaded China in December 2018 [9] and quickly spread to 27 provinces (autonomous regions and municipalities). In 2019, FAW occurred in nearly 1.2 million hm^2^ [10]. It was estimated that under the non-control scenario, the potential economic loss of maize caused by FAW would be USD 5.54 to 48.40 billion in China [11]. FAW has already established a seasonal migration pattern in China and breeds year round in some areas of southwestern China and southern China [12,13]. Because of the severe threat posed by FAW to China’s agricultural production and food safety, The Ministry of Agriculture and Rural Development of China added FAW to *The list of First-class Crop Diseases and Pests in China* [14].

Among insects, speciation mainly involves the evolution of distinct populations due to local adaptation to certain host plants [15,16]. FAW has a wide range of hosts, including more than 350 species in 76 families [4]. During its long-term evolution and spreading, FAW has differentiated into two strains associated with its main host plants: the maize strain, which prefers maize, sorghum and cotton and the rice strain, which prefers rice and pasture grasses [12,17,18]. Some studies have speculated that the FAW population that invaded China may be a special ‘maize strain’ descended from a hybrid population of rice-strain females and maize-strain males [19,20]. The host range of this particular strain may be further expanded during the invasion adaptation process [21].

Due to the differences in tissue structure, nutritional composition and secondary metabolites, host plants significantly affect insect growth, development, survival and reproduction [22,23]. Pests, therefore, have different preferences for different hosts [24,25]. Assessing the preferences of pests for different plants can help with predicting the occurrence of pests in different farmland ecosystems and controlling pests through the ecological regulation of regional crop arrangement. Su et al. [26] assessed the suitability of FAW to eleven kinds of plants, including five crops and six weeds in the lab, and found that maize and crabgrass were the most suitable hosts for FAW; they speculated that the FAW could shift from maize to surrounding weeds, especially crabgrass, in the field. Xiao et al. [11] studied the fitness of FAW on four vegetables and indicated that although the fitness of larvae and adults on vegetables was significantly lower than maize, the FAW could still establish a population on cucumber, tomato, cowpea, and Chinese cabbage; they speculated that FAW posed a potential threat to vegetable production.

Intercropping systems are environmentally friendly and sustainable agricultural production systems in which two or more crop varieties are grown simultaneously in the same field [27,28,29]. Such cultivation systems that increase plant diversity in the field have been shown to reduce pest populations and are widely used for pest control [30,31]. Certain crops and their arrangements will help disrupt host location by pests and act as repellents or deterrents, reducing oviposition on host crops [32,33]. Studies on the effects of intercropping on FAW have primarily focused on maize-based intercropping systems. Several studies have also documented that maize–legume intercropping can reduce the damage of FAW [34,35,36]. Furthermore, intercropping with edible legumes, such as groundnut and common bean, can also reduce FAW infestations considerably on maize [34]. Field studies in Uganda indicated that the infestation levels of FAW in maize fields intercropped with edible legumes were significantly lower than those of monocropped maize [34]. A comparative experiment from India on the FAW control of maize intercropping with five different legume crops also showed that maize–legume intercropping has a definite effect on reducing FAW damage in maize [35]. In China, Guo et al. [36] reported that although there was no significant effect of maize–soybean intercropping on the dispersal pattern of FAW, the larval survival rate was significantly lower than that of monocropped maize. However, these field experiments failed to explain the mechanism based on the biological characteristics of FAW. An indoor study that focused on a wheat-based intercropping system revealed FAW’s preference for wheat and its four common intercropping host plants and suggested that faba bean and maize were promising as push–pull plants to control FAW in wheat intercropping [37]. 

Southwestern China is one of the main maize-growing areas in China, with an annual maize-planting area of 15 percent of the country’s cultivated area. FAW can breed year round in some parts of southwestern China [38]. Monocropped maize systems and maize-based intercropping systems, such as maize–sweet potato, maize–peanut and maize–soybean intercropping, are widely adopted in southwestern China. The author’s previous research showed that, compared with the monocropped maize system, the above three maize-based intercropping systems could increase the arthropod diversity and reduce the population density of *Pyrausta nubilalis* (Hubern) and *Mythimna separata* (Walker) in the field. Among the three intercropping systems, the maize–sweet potato intercropping system was the most effective intercropping pattern for reducing the population density of *P. nubilalis* and *M. separata* [39]. Results from Liu et al. [40] showed that maize–peanut intercropping improved insect biodiversity and significantly reduced pest abundance. However, the effects of the maize–sweet potato and maize–peanut intercropping systems on FAW has not been reported in China. 

This study aims to elucidate the effect of maize-based intercropping host plants, including maize, sweet potato, soybean and peanut, on the growth, development and selection behavior of FAW. This study provides a reasonable explanation for how maize–legume intercropping reduces the damage caused by FAW and also discusses the potential roles of the maize–sweet potato/soybean intercropping in controlling FAW. 

## 2. Materials and Methods

### 2.1. Insects

The FAW was obtained from a laboratory colony maintained at Sichuan Agricultural University, Chengdu, China. The initial population was collected from maize fields in Luzhou City, Sichuan Province, China in June 2019. Larvae were fed on a regular artificial diet (Appendix A) and reared individually in finger-shaped tubes from the third instar to avoid cannibalism. Adults were fed on a 10% honey–water solution as a nutritional supplement. These insects were continuously reared in the laboratory under constant environmental conditions of 27 ± 1 °C, 60 ± 5% relative humidity and a L16:D8 photoperiod. All experiments were also carried out under the above environmental conditions.

### 2.2. Plants

The maize variety was “Colorful Glutinous Prince”. The soybean variety was “Liao Fresh One”, and the peanuts and sweet potatoes were local varieties. All of the aforementioned plants were planted in square plastic pots containing nutrient soil (Hogerrit Company, Chengdu, China). The cultivars were planted at varied intervals to ensure the availability of fresh leaves for the experiment. In addition, the host plants were planted in black plastic pots (9.3 cm high, 17.3 cm diameter at the mouth of the pot, 1500 mL capacity) according to the requirements of the oviposition selection test. All plants were incubated at a room temperature of 27 ± 1 °C, 60 ± 5% relative humidity and 16 h light: 8 h dark cycles.

### 2.3. The Development and Survival of FAW Fed on Different Host Plants

The development and survival of the FAW F_0_ and F_1_ generations fed on maize, soybean, peanuts and sweet potato were investigated and compared. Newly hatched FAW larvae were reared individually in plastic finger tubes (single head, single tube feeding) and fed separately on leaves of maize, sweet potato, peanut and soybean, and the plants were replaced daily to ensure sufficiency and freshness. Following adult emergence, the newly emerged females were randomly single-paired with concurrent males reared with the same plant and were fed on a 10% honey–water solution as a nutritional supplement. After the F_1_ generation eggs hatched, the larvae were reared in the same way as the F_0_ generation until the adults died, with 50 biological replicates of the F_0_ and F_1_ generations of each host plant fed. Growth and development were observed daily and recorded at all ages and survival rates.

### 2.4. Feeding Preference of FAW Larvae for Different Host Plants

Choice tests were conducted to determine the feeding preference of larvae between maize and another plant [37,41]. Three plant groups, namely, maize vs. soybean, maize vs. sweet potato and maize vs. peanut were compared. Each plant’s leaves were cut into leaf discs (2 cm in diameter). The 15 cm diameter petri dishes with moist square filter paper (8 cm side length) in the center of the bottom were used for the experiment. Two pieces of leaf discs of each host plant were alternated at the diagonal edge of the petri dish, so that the distance between each leaf disc and the center of the dish was equal. Ten larvae reared on an artificial diet were introduced into the center of the petri dish, and then the petri dish was sealed with sealing film to prevent the larvae from escaping. The number of larvae on each leaf disc was recorded after 6 and 24 h, respectively. Five replicates were performed for each group, and a total of 50 larvae were tested. The relative rate of larval feeding preference was calculated using the formula below. The feeding preference of the first instar larvae, the third instar larvae and the fifth instar larvae for different plants were determined.
$$\mathrm{Relative}\;\mathrm{feeding}\;\mathrm{preference}\;\mathrm{rate}=\frac{\mathrm{Number}\;\mathrm{of}\;\mathrm{larvae}\;\mathrm{on}\;\mathrm{the}\;\mathrm{host}\;\mathrm{plant}}{\mathrm{Number}\;\mathrm{of}\;\mathrm{total}\;\mathrm{larvae}}*100\%$$

### 2.5. Olfactory Responses of FAW Adults to Different Host Plants

The olfactory responses of FAW adults to different plants were performed on a Y-shaped tube olfactometer according to a published method [42]. The olfactometer consisted of a stem with an inner diameter of 2 cm and a length of 15 cm combined with two 10 cm long arms at an angle of 75°. Each arm was connected through plastic tubes to a 250 mL bottle as an odor source containing the leaves of the tested plant. An airflow of 400 mL min^−1^, which was previously filtered with water and activated charcoal, was passed through each olfactometer arm by the air pump. The airflow was controlled using a flow meter. The plant leaves were put into the source bottle. Fresh leaves from different host plants were weighed using an electronic analytical balance to ensure that the mass of leaves placed in the source bottle was the same. The adult was placed at the opening of the main arm and had 5 min to make a choice. The choice was considered valid if the moth flew into one arm and stayed there for at least 30 s. Otherwise, the response was recorded as invalid. Three kinds of groups were set up in the experiment: (1) the host plant vs. air (i.e., maize vs. air, soybean vs. air, peanuts vs. air and sweet potato vs. air), (2) maize vs. the other intercropping plant (i.e., maize vs. soybean, maize vs. peanuts and maize vs. sweet potato) and (3) the combination of maize and intercropping plant vs. maize (i.e., soybean + maize vs. maize, peanuts + maize vs. maize, and sweet potato + maize vs. maize). In the third kind of group, each component of the combination accounted for half the mass of the combination. A total of 30 adults (15 males and 15 females) for each group were tested individually. To avoid bias caused by light, airflow, etc., the arms of the olfactometer were replaced after every five insects, and the olfactometer was wiped with alcohol and air dried when switching plant species. The relative olfactory selection rate of the adult was calculated using the formula below.
$$\mathrm{Relative}\;\mathrm{olfactory}\;\mathrm{selection}\;\mathrm{rate}=\frac{\mathrm{Number}\;\mathrm{of}\;\mathrm{insects}\;\mathrm{that}\;\mathrm{selected}\;\mathrm{the}\;\mathrm{host}\;\mathrm{plant}}{\mathrm{Total}\;\mathrm{adults}\;\mathrm{tested}}*100\%$$

### 2.6. Oviposition Preference of Adult FAW for Different Host Plants

The oviposition preference of adult FAW was measured in the choice tests. Two types of groups were set up in the experiment: (1) the intercropping plant vs. maize (i.e., soybean vs. maize, peanuts vs. maize and sweet potato vs. maize), (2) the combination of maize and intercropping plant vs. maize (i.e., soybean + maize vs. maize, peanuts + maize vs. maize and sweet potato + maize vs. maize). In the first type, one pot with one seeding of each plant in the group was arranged in a mesh cage (40 × 35 × 50 cm), and five replicates were performed for each group. In the second type, the combination plant pot had one seedling of each tested crop (for example, the combination of maize + soybean had one seeding of maize and one seeding of soybean), while the maize pot had two seedings of maize. Four replicates were performed for each group. All host plants were planted at the same time and tested after about 21 days. Five pairs of unmated 2-day-old adult FAW were introduced into the center of the cage. A 10% honey–water solution was hung on the top center of the cage as a nutritional supplement. The potted plants were changed, and the egg masses and egg numbers were recorded on each plant every day until the adult stopped laying eggs. The egg masses laid on the cage surface were discarded. 

### 2.7. Statistical Data Analysis

All statistical analyses were performed using SPSS 27.0 software (SPSS Inc., Chicago, IL, USA). The Shapiro–Wilk and Levene’s tests were used to verify the assumptions of normality and homogeneity of variance before analysis. The nonparametric Kruskal–Wallis test of variance using multiple comparison tests and Mann–Whitney U test were used to estimate the effect of different host plants on the growth and development of FAW. The chi-square test was used to estimate the feeding preference of FAW larvae and the olfactory selection of FAW adults. The Wilcoxon matched-pairs signed-ranks test or paired t-test was used to estimate the oviposition preference of FAW adults. A *p* value < 0.05 was considered statistically significant.

## 3. Results

### 3.1. Effect of Different Host Plants on the Growth and Development of FAW

As shown in Table 1, the F_0_ generation had the highest average survival, pupation and emergence rates of larvae fed on maize at 84%, 82% and 82%, respectively. The larvae fed on sweet potato had the lowest average survival, pupation and emergence rates at 12%, 6% and 6%, respectively, and only the female adults emerged successfully. Therefore, the growth and development of the F_1_ generation fed on sweet potato was not observed. There were no significant differences in egg duration among the FAW fed on the four host plants. However, differences in the larval duration were significant. The larval duration of FAW fed on maize was the shortest (17 d) and was 4, 5.5 and 7 days shorter than the larval duration of FAW fed on peanut, sweet potato and soybean, respectively. The pupal duration fed on soybean (8 d) was shorter than that of the other three host plants, whereas the longest pupal duration was observed on sweet potato (10 d), but it was not significantly different from maize (9 d) and peanut (9 d). The longevity of FAW female adults fed on soybean was the longest (11 d). Since none of the male adults fed on sweet potato emerged successfully and could not mate, the preoviposition duration was unknown, and the generation duration of FAW fed on sweet potato was unavailable. Significant differences were observed in the generation duration of FAW fed on maize, peanut and soybean. The FAW fed on maize had the shortest generation duration, and the FAW fed on soybean had the longest. 

Similarly, the larval survival, pupation and emergence rates of the F_1_ generation fed on maize were the highest at 82%, 80% and 80%, respectively. The larvae fed on soybean had the lowest survival, pupation and emergence rates at 14%, 8% and 8%, respectively, and only a few females emerged (Table 2). The host plants also had a significant effect on the development of the FAW F_1_ generation. Compared to maize, the larval duration of FAW fed on soybean and peanut was significantly longer by 11 d and 6.5 d, respectively, and the pupal duration was significantly longer by 2.5 d and 1 d, respectively. The longevity of female adults fed on soybean was significantly shorter than those fed on maize by 4 d, while the longevity of adult females fed on peanut was not significantly different from those fed on maize. Only female adults of the F_1_ generation fed on soybeans successfully emerged, so the longevity of male adults and the generation duration of the F_1_ generation fed on soybean was not available. This result indicated that maize and peanut were suitable for the survival and development of FAW, while sweet potato and soybean were not suitable for the multigenerational reproduction of FAW.

### 3.2. Feeding Preference of FAW Larvae for Maize and Other Three Plants

After 6 h, the feeding preference of the first instar larvae for sweet potato (χ2 = 0.03, *df* = 1, *p* = 0.86) and peanut (*χ*2 = 0.67, *df* = 1, *p* = 0.41) did not differ significantly from that of maize, while soybean (*χ*2 = 14.88, *df* = 1, *p* < 0.001) was significantly lower than maize. The third and fifth instar larvae showed a significantly lower feeding preference for sweet potato (third: *χ*2 = 10.52, *df* = 1, *p* = 0.001; fifth: *χ*2 = 7.04, *df* = 1, *p* = 0.008), peanut (third: *χ*2 = 8.81, *df* = 1, *p* = 0.003; fifth: *χ*2 = 10.80, *df* = 1, *p* = 0.001) and soybean (third: *χ*2 = 5.49, *df* = 1, *p* = 0.02; fifth: *χ*2 = 10.52, *df* = 1, *p* = 0.001) than maize (Figure 1A). After 24 h (Figure 1B), the first, third and fifth instar larvae significantly preferred to feed on maize compared to the peanut (first: *χ*2 = 5.77, *df* = 1, *p* = 0.02; third: *χ*2 = 4.12, *df* = 1, *p* = 0.04; fifth: *χ*2 = 11.76, *df* = 1, *p* < 0.001); sweet potato (first: *χ*2 = 4.59, *df* = 1, *p* = 0.03; third: *χ*2 = 6.40, *df* = 1, *p* = 0.01; fifth: *χ*2 = 4.79, *df* = 1, *p* = 0.03) and soybean (first: *χ*2 = 22.26, *df* = 1, *p* < 0.001; third: *χ*2 = 8.02, *df* = 1, *p* = 0.005; fifth: *χ*2 = 4.79, *df* = 1, *p* = 0.03).

### 3.3. Olfactory Selection of FAW Adults for Maize and Other Three Plants

When FAW adults selected between host plant and air, they showed a significantly higher selection for maize (*χ*2 = 9.14, *df* = 1, *p* = 0.003), sweet potato (*χ*2 = 6.76, *df* = 1, *p* = 0.01) and peanut (*χ*2 = 4.17 *df* = 1, *p* = 0.04) than air. In contrast, FAW showed a significantly lower selection for soybean (*χ*2 = 13.33, *df* = 1, *p < 0.001*) than air (Figure 2A). When FAW selected between the intercropping plant and maize, the preference of the FAW for maize was significantly higher than that of sweet potato (*χ*2 = 5.54, *df* = 1, *p* = 0.02) and soybean *(χ*2 = 15.21, *df* = 1, *p* < 0.001); it was also higher than for peanut (*χ*2 = 1.29, *df* = 1, *p* = 0.26), but the difference was not significant (Figure 2B). When FAW selected between the combination (maize + intercropping plant) and maize, the preference for maize was significantly higher than that for the maize + peanut (*χ*2 = 4.80, *df* = 1, *p* = 0.03) and the maize + soybean (*χ*2 = 10.70, *df* = 1, *p =* 0.001) combinations; however, there was not a significant difference between the preferences for maize and the maize + sweet potato combination (*χ*2 = 0.57, *df* = 1, *p* = 0.45) (Figure 2C). The result indicated that soybean showed a strong deterrent effect against FAW adults.

### 3.4. Oviposition Preference of FAW Adults for Maize and Other Three Plants

The oviposition preference of FAW for different plants were showed in Table 3. In the maize vs. intercropping plant tests, female adults preferred to oviposit on maize rather than on the intercropping plants. The egg masses and eggs on maize were more abundant than on the other three plants. In each test, the oviposition percentage of FAW on maize approached 100%. 

Similarly, the egg mass in the maize vs. (maize + sweet potato) test for maize was greater than the combination and the maize vs. (maize + soybean) test the egg masses and egg numbers for maize were greater than the combination. However, in the maize vs. maize + peanut test, there were no significant differences in the egg masses or egg numbers between maize and the combination. The oviposition percentages of FAW on maize in the maize vs. the combination test were between 49.67 and 63.52%, which was less than on maize in the maize vs. the single intercropping plant tests.

## 4. Discussion

Intercropping systems with an appropriate plant configuration can improve farmland ecological diversity, which is conducive to the proliferation of natural enemy species and numbers and, to some extent, reduces the pest population density and crop damage [43]. For example, the well-known and traditional “Milpa” cultivation from Central America has significant benefits for weed, pathogen and pest control, productivity and overall profitability [44]. The maize, sweet potato, soybean and peanut were the common host plants of maize-based intercropping systems in southwestern China, and it is important to investigate the effect of each configuration of host plants on the biological parameters of the growth, development and selection behaviour of FAW and to assess its ability to control pests in maize-based intercropping systems.

The development and survival of phytophagous insects are strongly affected by the host plant [45]. Different plant configurations in the intercropping systems have different effects on specific pest control abilities [46]. Generally, the host fitness for phytophagous insects is indicated by a high survival rate and a short developmental duration of the larval stage [47]. As with most studies [25,26,48], our findings also indicated that FAW fed on maize had the highest survival rate and shortest developmental duration, suggesting that maize was the most suitable host plant for FAW. This result is consistent with the fact that the FAW that invaded China may be a special ‘maize strain’ [19,20] and prefer to infest maize crops [49]. Compared with FAW fed on maize, FAW fed on peanut, soybean and sweet potato showed lower survival rates and slower development. Especially, only a few females of the F_0_ generation of FAW fed on sweet potato and the F_1_ generation fed on soybean successfully emerged and could not continue to reproduce in our study. This result is partially different from the findings of Xu et al. [50], which showed that FAW could complete their entire life cycle on soybean with stunted development, a reduced survival rate, a shortened adult stage and a decreased fecundity and oviposition duration compared with FAW fed on maize. However, Xu’s study focused on only one generation of FAW and did not consider the effect of the host on the multigenerational reproduction of FAW. It is important to note that peanuts extended the developmental duration of FAW larvae, but the survival rate was high, and the pest could complete its life cycle. Therefore, this study showed that among the four host plants, maize and peanut were suitable for the growth, development and reproduction of FAW, while the opposite was true for sweet potato and soybeans. However, the exact mechanism by which soybean and sweet potato cause low FAW survival, pupation and emerging rates and slow growth needs further investigation.

The larval feeding preference is one of the indicators used to evaluate the acceptance of hosts by insects [51]. However, previous diet and experience can modify insect feeding behavior. In order to examine this, Boiça Júnior et al. [52] conducted a study of the influence of the fall armyworm’s previous experience with soybean genotypes on larval feeding behavior. The results show that FAW does not show habituation and induction of preference toward the experienced soybean genotypes. They surmised that the habituation and induction of preference do not occur in FAW within a short period of larval experience. However, it is still unclear whether long-term previous diets can change the habituation and induction of preference of FAW. The FAW that invaded China was mainly of the “maize-strain” variety; therefore, we mainly fed the test FAW with artificial diets based on maize powder to reduce the effect of the previous diet on the experimental results. As we expected in our study, FAW larvae had a feeding preference for maize compared to other crops. This result is consistent with the findings of Li et al. [53] and Xu et al. [54]. However, the first instar larvae did not demonstrate a significant feeding preference in maize vs. sweet potato and maize vs. peanut treatment after a 6 h inoculation but did after a 24 h inoculation. Li et al. [55] also found that the feeding preference of early instar larvae for different host plants over a brief period of time was minor, whereas the feeding preference of old instar larvae was significant. A possible explanation is the difference in mobility between the early instar and the old instar larvae. If given a longer period of time, early instar larvae would select more suitable hosts. In the maize vs. soybean treatment, the first instars showed a strong preference for maize, which may be related to the fact that soybean had volatile substances that are repellent to FAW. This can also be illustrated using the results of the adult olfactory selection test in our study. Another explanation is that the physical structure of the soybean surface is significantly hairier compared to sweet potato and peanut, which may act as a physical barrier and as a trichome-based plant defense mechanism to inhibit the feeding activity of early instar larvae. 

Locating a host plant is crucial for a phytophagous insect to find suitable oviposition sites. According to the ‘preference–performance hypothesis’, also known as the ‘optimal oviposition theory’ [56] or ’mother knows best’ theory [57], females choose to oviposit on hosts where their larvae will thrive [58,59]. Our results indicated that maize odor had a significant attractive effect on FAW adults, while soybean odor had a strong deterrent effect on FAW adults. Furthermore, the intercropping plants reduced the selection rate of FAW compared to maize alone. The same results were seen in oviposition preference tests in our study in which FAW females had a significant oviposition preference for maize rather than for other plants but laid the least number of eggs on soybean. Plant volatiles, either species-specific compounds or specific ratios of ubiquitous compounds, play an important role in the host-location process at both the adult and larval stages [60,61,62]. On the other hand, some specific volatile compounds released by plants produce repellent effects against insects. Several studies have confirmed that some leguminous crops, such as *Desmodium intortum*, can emit semiochemicals, such as (E)-β-ocimene and (E)-4,8-dimethyl-1,3,7-nonatriene, that deter female oviposition on maize [63,64]. Liu et al. [65] also discovered that volatiles from resistant soybean leaves were repellent to *Trichoplusiani* and demonstrated that the major components were tetradecene and dodecene in comparison to the attracting action of the “Davis” soybean. However, the specific compounds in soybeans that repel FAW and its offspring need to be further elucidated.

Studies have demonstrated that maize–soybean intercropping reduces FAW infestation in the field compared to a monocropped maize system [34,36]. Our study indicated that soybean reduced FAW larval fitness and had a strong deterrent effect on the host location and oviposition of FAW. These may partially explain the mechanisms of maize–soybean intercropping that reduce FAW damage in the field. Maize–soybean intercropping, which is an effective approach to improve crop yield and nutrient use efficiency, is widely practiced by farmers and strongly promoted by the government in China in recent years [66,67]. We speculate that the extension of this intercropping pattern will reduce the occurrence of FAW and the damage caused by FAW in China. In addition, our study also found that, similar to soybean, sweet potato reduced the development and reproduction of FAW and interfered with the oviposition selection of FAW. Our previous field investigation also indicated that the maize–sweet potato intercropping system was the most effective intercropping system for reducing the population density of the other two major pests of maize, *P. nubilalis* and *M. separata*. Therefore, we have reason to believe that maize–sweet potato intercropping may also have the potential to control FAW, while the peanut was assumed to have weak pest control effects on FAW. However, this needs to be confirmed through field investigations.

A push–pull strategy is a companion cropping system that includes exploiting behavior-modifying stimuli to manipulate the distribution and abundance of pests and their natural enemies [68]. Plants that have a strong attractive effect on egg-laying females were selected as trap crops (pull) to draw pests away from the main crops, while plants that have a highly repellent effect on females were selected as intercrops and employed as repellent plants (push) in the intercropping system for pest control [33,69]. Regarding this practice in Africa and Mexico [70], some plants, such as Brachiaria cv. Mulato II, Napier grass and Pennisetum purpureum Schumach, were successfully used as trap plants. Therefore, we suggest that planting these trap plants around an intercropped field with maize–soybean/sweet potato intercropping may reduce the FAW infestation. Furthermore, soybeans and sweet potatoes are highly complementary; the soybeans and sweet potatoes shade the ground, reducing bare soil disincentives, weed growth and propitiates based on the total productivity of the maize intercropping system [44]. Therefore, maize–soybean/sweet potato intercropping is worthy of further study in the prevention and control of FAW.

## 5. Conclusions

Overall, our results suggest that FAW fail to complete multiple generations in sweet potato and soybean, which is detrimental to their growth and development. FAW also displayed a low preference for soybean and sweet potato. Furthermore, the maize–sweet potato and maize–soybean configurations could reduce FAW infestations. Soybean and sweet potato have potential value in preventing FAW in intercropping systems based on maize, and their specific field defense mechanisms need further research.

## Figures and Tables

**Figure 1 insects-15-00026-f001:**
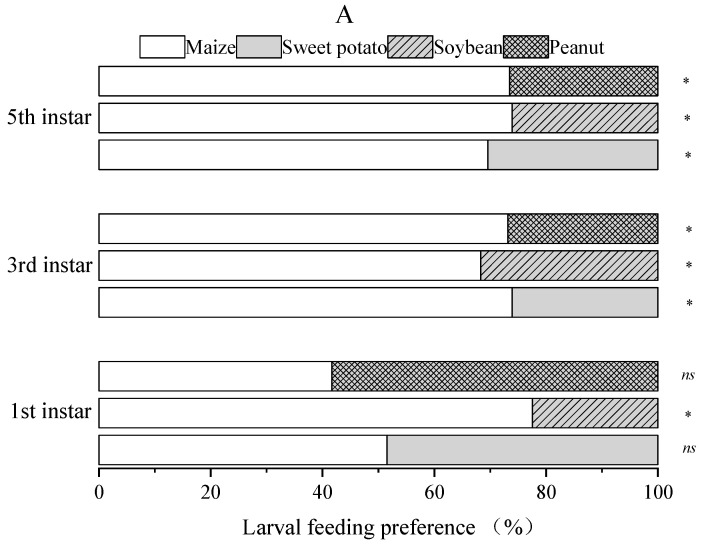
Feeding preference of the first instar, third instar and fifth instar FAW larvae for four different plants. Note: (**A**) 6 h and (**B**) 24 h; * indicates significant differences (chi-square test, *p* < 0.05); and *^ns^* indicates no significant differences (chi-square test, *p* > 0.05). A chi-square test was conducted using the actual number of larvae selected.

**Figure 2 insects-15-00026-f002:**
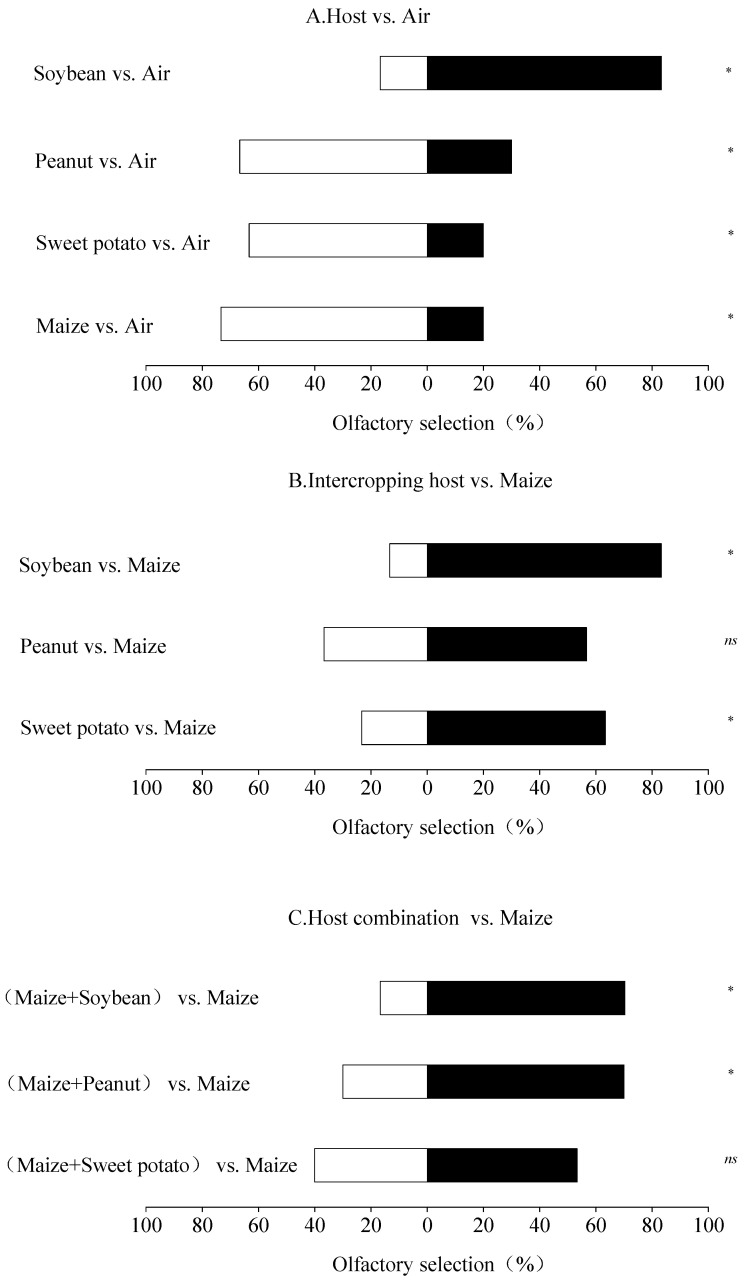
Olfactory selection of FAW adults for maize and the other three plants (**A**–**C**). Note: The values in the horizontal coordinates of the graph are the percentage of selection; * indicates significant differences (chi-square test, *p* < 0.05); and *^ns^* indicates no significant differences (chi-square test, *p* > 0.05). A chi-square test was conducted using the actual number of adults selected.

**Table 1 insects-15-00026-t001:** Developmental duration of FAW F_0_ generation fed on four different plants.

Stage	Developmental Duration (Days)				*χ*2	*p*
	Maize	Sweet Potato	Peanut	Soybean		
Egg	2 (0) a	2 (0) a	2 (0) a	2 (0) a	3.72	0.29
First instar larva	2 (1) c	3 (1) a	2 (0) d	3 (0) b	107.91	<0.001
Second instar larva	2 (1) d	2 (0) c	2 (1) b	3 (1) a	70.79	<0.001
Third instar larva	2 (0.25) b	2 (1) a	3 (1) a	2 (1) a	23.63	<0.001
Fourth instar larva	2 (1) b	3.5 (1) a	3 (2) a	3 (1) a	21.24	<0.001
Fifth instar larva	3 (1) b	4 (1.5) a	4 (0) a	4 (1) a	43.83	<0.001
Sixth instar larva	6 (1) c	8 (1) a	7 (1.5) b	7.5 (1) a	37.82	<0.001
Total larval duration	17 (2.25) c	22.5 (2) a	21 (2) b	24 (1) a	65.75	<0.001
Pupa	9 (2) a	10 (0) a	9 (2) a	8 (1) a	7.47	0.06
Pre-adult duration	28 (2.5) b	34 (0) a	32 (3) a	34 (1.25) a	53.32	<0.001
Female adult	7 (1) b	7 (0) ab	9 (4) ab	11 (1) a	8.64	0.04
Male adult	6 (2) b	—	6.5 (2.5) b	12 (0) a	9.45	0.009
Preoviposition duration	3 (1) a	—	3 (1.5) a	4 (1.5) a	4.46	0.11
Generation	29 (1.25) b	—	34 (4.5) a	37 (3) a	19.9	<0.001
Larval survival rate % (No.)	84 (42)	12 (6)	74 (37)	24 (12)	—	—
Pupation rate % (No.)	82 (41)	6 (3)	72 (36)	20 (10)	—	—
Emergence rate % (No.)	82 (41)	6 (3)	72 (36)	20 (10)	—	—

Note: The values in the table are presented as the median (interquartile range, Q3–Q1). χ2 and *p* represent the statistic and significance, respectively, of the Kruskal–Wallis test. Different lowercase letters after data in the same row indicate significant differences at the 0.05 level in Kruskal–Wallis post hoc comparisons.

**Table 2 insects-15-00026-t002:** Developmental duration of F_1_ generation of FAW fed on three different plants.

Stage	Developmental Duration (Days)			*χ*2/Z	*p*
	Maize	Peanut	Soybean		
Egg	2 (0) b	3 (0) a	3 (0) a	135.98	<0.001
First instar larva	3 (1) b	4 (0.75) a	3 (1) b	67.84	<0.001
Second instar larva	2 (0) b	3 (1) a	2 (1) b	67.84	<0.001
Third instar larva	2 (1) c	5 (2) a	3 (1) b	36.64	<0.001
Fourth instar larva	3 (1) b	4 (3) a	3 (1.5) ab	13.69	0.001
Fifth instar larva	4 (2) b	4 (1) b	12.5 (9.75) a	15.35	<0.001
Sixth instar larva	6 (1) c	8 (1.25) b	9 (0) a	35.13	<0.001
Total larval duration	21 (2) c	27.5 (3) b	32 (5) a	66.3	<0.001
Pupa	9 (2) a	10 (1) a	11.5 (3.25) a	24.25	>0.001
Pre-adult duration	33 (2) c	39 (2) b	46 (2) a	60.13	<0.001
Female adult	10.5 (2.25) a	11 (6) a	6.5 (1) b	6.29	0.04
Male adult	9 (4) *	11.5 (4)	—	179.5	0.04
Preoviposition duration	3 (1.25)	3 (1)	—	109.5	0.30
Generation	35 (2) *	43.5 (1.25)	—	180	<0.001
Larval survival rate % (No.)	82 (41)	76 (38)	14 (7)	—	—
Pupation rate % (No.)	80 (40)	70 (35)	8 (4)	—	—
Emergence rate % (No.)	80 (40)	66 (33)	8 (4)	—	—

Note: The values in the table are presented as the median (interquartile range, Q3–Q1). χ2/Z and *p* represent the statistic and significance, respectively, of the Kruskal–Wallis test/Mann–Whitney U test. Different lowercase letters and * after data in the same row indicate significant differences at the 0.05 level in Kruskal–Wallis post hoc comparisons and the Mann–Whitney U test, respectively.

**Table 3 insects-15-00026-t003:** Oviposition preference of FAW adults for maize and other three plants.

Host Plant	Egg Masses	Eggs	Oviposition Percentage (%)
Maize	12 (3.5) *	1322 (508) *	100 (3.85) *
Sweet potato	0 (0.5)	0 (53.5)	0 (3.84)
Z	−2.07	−2.02	−2.06
*p*	0.04	0.04	0.04
Maize	14.60 ± 3.53 *	1040 (623) *	95.27 ± 2.34 *
Peanut	0.80 ± 0.04	33 (168.5)	4.73 ± 2.34
Z/t	4.25	−2.02	19.36
*p*	0.01	0.04	<0.001
Maize	13 (6.5) *	1231 (391.5) *	100 (4.33) *
Soybean	0 (1)	0 (40)	0 (3.33)
Z	−2.02	−2.02	−2.04
*p*	0.04	0.04	0.04
Maize	18.50 ± 0.96 *	1531.67 ± 195.63	63.27 ± 3.45 *
Maize + Sweet potato	9.00 ± 1.96	687.33 ± 152.83	36.73 ± 3.45
t	7.98	3.28	3.85
*p*	0.004	0.08	0.03
Maize	12.33 ± 2.40	1316.67 ± 107.80	49.67 ± 5.85
Maize + Peanut	15.00 ± 1.73	1171.33 ± 223.43	50.33 ± 5.85
t	−1.6	0.55	−0.05
*p*	0.25	0.64	0.96
Maize	21.00 ± 1.73 *	1067.00 ± 111.50 *	65.33 ± 4.164 *
Maize + Soybean	8.33 ± 1.38	608.00 ± 51.57	34.67 ± 4.164
t	4.75	7.42	6.38
*p*	0.04	0.02	0.02

Note: If the data followed a normal distribution and had homogeneity of variance, then they were presented as the mean ± SE; otherwise, the median (interquartile range, Q3–Q1) was used. The * symbol indicates a paired t-test or Wilcoxon matched-pairs signed-ranks test for significant differences at the 0.05 level. t/Z and *p* represent the paired *t*-test/Wilcoxon matched-pairs signed-ranks test statistic and significance of differences, respectively.

## Data Availability

The data presented in this study are available on request from the corresponding author.

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
