# Peer review of "Effects of the Host Plants of the Maize-Based Intercropping Systems on the Growth, Development and Preference of Fall Armyworm, Spodoptera frugiperda (Lepidoptera: Noctuidae)"

_insects, 2024, doi:10.3390/insects15010026_

Round 1

Reviewer 1 Report

Comments and Suggestions for Authors

Tao et al. report the results of laboratory experiments on fall armyworm host plant selection and immature development. Their results could have applications in the selection of combinations of plants for intercropped maize production systems.

Main points

I was concerned that some of the results did not appear to meet the assumptions of equality of variances that is necessary for ANOVA. It seems likely that non-parametric tests may be necessary to analyze these data. In the case of t-tests, heteroscedasticity can be handled by applying Welch's adjustment (Welch's t-test). The authors need to check the variance and normality of their data prior to analysis.

Discussion. The Discussion should include the following issues: (i) How previous diet (e.g. artificial diet) can affect larval feeding behavior of Spodoptera species (there are several studies on this); (ii) The long history of traditional use maize-bean or maize-bean-squash cultivation in Central America – this is known as the "Milpa" and has been practiced for hundreds of years. (iii) What effects do other plants have in maize crops, such as weeds?

I have written numerous suggestions and numbered points on a scanned copy of the manuscript

Numbered points (see scanned manuscript)

1. I did not understand the phrase "scientific control". Please reword.

2. The Abstract is too long. Almost 50% is preamble. Please focus on your results and reduce length by 40%.

3. But did this concern develop into a serious problem? Is FAW a serious pest or just a minor issue in vegetable production in China?

4a. Population "regulation" involves density-dependent mortality. You mean "reduce".

4b. Diet can strongly affect larval feeding responses, so please provide details of the artificial diet (a list of ingredients and quantities as Supplemental Material would be useful).

5. Were these experiments performed in cages?

6. As you only used 50 insects, how many "families" did these larvae originate from? There could be strong parental effects here, so having a variety of parents would be important.

7. Did 1st, 3rd or 5th instars consume artificial diet before this experiment?  Prior diet has a DRAMATIC effect on insect feeding behavior in Spodoptera species.

8. How did you establish that the "quality" of leaves was the same in each sample?

9. Were seedlings of similar size, or what range of sizes of plants were tested?

10. Normal distribution and equality of variances is an prerequisite for ANOVA and for t-test. How did you checked that your data met these assumptions? I saw some concerning issues in your results.

11. Please indicate what these series of percentages refer to....better to say "respectively" in each case.

12. Table 1. No need to present percentages to two decimal places as you only used 50 insects! e.g. "84%" or "70%" would be sufficient.

13. Table 1. Is there any value in comparing the frequencies among host plants by contingency table (Chi2 values) to provide a statistic support for the observed variation in survival, pupation, etc.? You may want to discuss this with a statistician.

13 (lines 236 in text). Again you are reporting larval duration to two decimal places when you sample size was not large. Suggest you report all duration values (egg, larvae, pupae, generation) to one decimal place throughout the manuscript (including Table 2, Table 3).

14. Table 2. Results for First instar, 6th instar, female adult, and male adult values appear to have unequal variances. Please test for homoscedasticity, and if necessary perform non-parametric tests.

15. Table 3. Sixth instar, total larval duration and Pupal duration values appear to have unequal variances. See point 14.

16. I did not understand this sentence. Please clarify. Selection rate of what?

17. Delete first column for Table 4, as this information appears in column 2.

18. Table 4. Is "1.00" a mean? where is the SE value? Similarly for 107.00.

19. Table 4. All these values appear to have unequal variances. See point 14.  As you performed t-tests, you may need to perform Welch's correction (Welch's t-test) to account for these inequalities.

20. I did not understand this sentence. Please clarify. Do you mean the specific mechanisms by which sweet potato and soybean delay larval development?

Comments on the Quality of English Language

Minor editing (see my suggestions).

Author Response

Dear reviewer of Insects,

Thank you for your constructive comments concerning our manuscript entitled “Effects of the host plants of the maize-based intercropping systems on the growth, development and preference of Spodoptera frugiperda (Smith) (insect -2714699)”. We have studied your comments carefully and made major rectification which we hope to meet with your approval. We answer your questions or comments in details in the annex.

Reviewer 2 Report

Comments and Suggestions for Authors

General comments

I have read the manuscript (insects-2714699). Entitle: Effects of the host plants of the maize-based intercropping systems on the growth, development, and preference of Spodoptera frugiperda (Smith), written by Tao et al. for publication in Insects/MDPI. They investigated the effects of the host plant of the maize-based intercropping systems, including maize, sweet potato, soybean, and peanut, on the growth, development, and preference of the fall armyworm in the laboratory. The overall finding from this study is interesting. However, the author needs to provide all the statistical values clearly in the results section.

Overall, after I evaluate and request the author for this manuscript as a “MAJOR REVISION”.

Title: Please include the common name and order for FAW in the title.

Introduction:

Line 71-73: There are several studies on the effect of host plants on the development, survival, and reproduction of fall armyworm. So please cite the paper, which is related to the fall armyworm rather than other insects. For example:

1.     Acharya et al., 2022. Impact of Rice and Potato Host Plants Is Higher on the Reproduction than Growth of Corn Strain Fall Armyworm, Spodoptera frugiperda (Lepidoptera: Noctuidae)  https://doi.org/10.3390/insects13030256

2.     Lu et al., Preference and performance of the fall armyworm, Spodoptera frugiperda, on six cereal crop species  https://doi.org/10.1111/eea.13307

Line 93: groundnut soya and common bean, can also reduce FAW infestations, please check it.

Line 102-103: Indoor study for the host preference of FAW may give us a clue: Remove it.

Line 117-118: However, few studies have focused on the effects of the 117 maize-sweet potato, and maize-peanut intercropping systems on FAW, please provide the reference for this statement.

Materials and methods:

Line 128-129: Larvae were fed on a regular artificial diet, please provide the details of artificial diet.

Line 132: 27 ± 1℃, 60± 5%RH, please check the space between number and symbol and for RH.

Line 135-136: Please provide the cultivars of all crops used for this study.

Line 137: What is nutrient soil, please provide in details.

Line 142: relative air humidity, write only relative humidity.

Line 153-154: Please rephrase this sentence. 

Results:

Provide the statistical value for all significantly difference results including p value, df and F value.

Table 1: Please provide the mean comparison value with p value.

Table 2: Were the larval developmental stages for all host plants same? Provide the preadult duration, preadult survival, pre-oviposition days in the table. Same for table 3.

Please provide the conclusion of your study.  Please provide the key messages based on your research outcomes only. I would love to read striking points and take-home messages that will linger in the readers’ minds. What is the novelty, how does the study elucidate some questions in this field, and the contributions the paper may offer to the scientific community?

References: Please double-check the citations, their style, spell check, and other grammatical errors.

Good Luck!

Author Response

Dear reviewers of Insects,

Thank you for your time and effort in reviewing our manuscript. We are grateful for your constructive feedback and thoughtful suggestions. Not only have you provided valuable guidance on the overall structure and content of the article, but you have also pointed out many helpful details that we had overlooked. We are now making changes based on your comments and answering your questions line by line. See the appendix for details.

Round 2

Reviewer 1 Report

Comments and Suggestions for Authors

The authors have addressed my concerns.

Comments on the Quality of English Language

Requires editing during journal production.

Author Response

Dear reviewer of Insects

Thank you again for your valuable comments on our manuscript! This time, we have made modifications and responses according to the requirements of Reviewer 2. Please see the attachment for details.

Reviewer 2 Report

Comments and Suggestions for Authors

I can tell that the authors put lots of effort into improving this manuscript. However, there are some errors that should be addressed before it is suitable for publication.

Please check the statistical analysis part, particularly the t value provided in the results section. I believe the value should come from the chi-square test. This is most important.

Please rewrite the conclusion part more precisely.

Line 142:  and soybean.and the plants were, Please remove "." after soybean.

Good Luck!

Author Response

Dear reviewer of Insects,

Thank you for your comments on our manuscript. We have made the necessary changes and provided a response as requested. Please see the attached document for details.

Round 3

Reviewer 2 Report

Comments and Suggestions for Authors

The authors have improved the manuscript according to the comments and suggestions of reviewers and replied point-by-point to the reviewers' questions.  Good Luck!